# Destroying the Shield of Cancer Stem Cells: Natural Compounds as Promising Players in Cancer Therapy

**DOI:** 10.3390/jcm11236996

**Published:** 2022-11-26

**Authors:** Melania Lo Iacono, Miriam Gaggianesi, Paola Bianca, Ornella Roberta Brancato, Giampaolo Muratore, Chiara Modica, Narges Roozafzay, Kimiya Shams, Lorenzo Colarossi, Cristina Colarossi, Lorenzo Memeo, Alice Turdo, Veronica Veschi, Simone Di Franco, Matilde Todaro, Giorgio Stassi

**Affiliations:** 1Department of Health Promotion Sciences, Internal Medicine and Medical Specialties (PROMISE), University of Palermo, 90127 Palermo, Italy; 2Department of Surgical, Oncological and Stomatological Sciences (DICHIRONS), University of Palermo, 90127 Palermo, Italy; 3Department of Experimental Oncology, Mediterranean Institute of Oncology, Viagrande, 95029 Catania, Italy

**Keywords:** natural products, cancer stem cells, drug resistance, alkaloids, flavonoids, polyphenols, adjuvant treatments

## Abstract

In a scenario where eco-sustainability and a reduction in chemotherapeutic drug waste are certainly a prerogative to safeguard the biosphere, the use of natural products (NPs) represents an alternative therapeutic approach to counteract cancer diseases. The presence of a heterogeneous cancer stem cell (CSC) population within a tumor bulk is related to disease recurrence and therapy resistance. For this reason, CSC targeting presents a promising strategy for hampering cancer recurrence. Increasing evidence shows that NPs can inhibit crucial signaling pathways involved in the maintenance of CSC stemness and sensitize CSCs to standard chemotherapeutic treatments. Moreover, their limited toxicity and low costs for large-scale production could accelerate the use of NPs in clinical settings. In this review, we will summarize the most relevant studies regarding the effects of NPs derived from major natural sources, e.g., food, botanical, and marine species, on CSCs, elucidating their use in pre-clinical and clinical studies.

## 1. Introduction

Despite prominent advances in the field of cancer prevention and early diagnosis, it is expected that one in five people will develop cancer during their lifespan. One of the greatest challenges in translational oncology is limiting the onset of primary or acquired drug resistance, which is boosted by cancer stem cells (CSCs). CSCs represent a pluripotent heterogeneous population within tumor bulk, with self-renewal and differentiation abilities, contributing to the failure of conventional therapies and, therefore, to disease relapse and metastasis [1]. Increasing evidence points out that different natural products (NPs) can modulate the CSCs’ hallmarks and sensitize them to conventional treatment [2]. NPs show minimal side effects in comparison with chemotherapeutics, and many studies have demonstrated their emerging role as adjuvant agents in cancer treatment. In this review, we point out the potential effects of major NPs derived from different origins (dietary, botanical, and marine sources) on CSCs in pre-clinical and clinical settings.

## 2. Cancer Stem Cells: The Main Players in Drug Resistance

Drug resistance is doubtless the main challenge of treatment in cancer patients. It is possible to distinguish two categories of drug resistance: intrinsic resistance and acquired resistance after drug treatment [3]. Compelling evidence highlights that intratumoral heterogeneity is one of the major hurdles involved in intrinsic drug resistance, in which the CSCs represent the main players due to their self-renewal and differentiation abilities [4,5]. The presence of CSCs has been characterized in different tumors, such as thyroid, colorectal, breast, prostate, and other solid tumors [6,7]. CSCs are identified and can be isolated by the expression of specific surface markers such as CD133 [8], CD44 [9], CD44v6 [10], EpCAM [11], or enzyme activity such us ALDH [12]. CSCs are also defined as tumor-initiating cells, as they can generate tumor xenografts in immunocompromised mice models [13]. Moreover, the failure of conventional therapies, based on the use of radiotherapy and chemotherapy to induce DNA damage in highly proliferative cells and to eradicate tumor mass, is strictly due to the presence of CSCs, which are characterized by multiple survival mechanisms [14]. In particular, the mechanisms through which CSCs escape chemotherapeutic treatments are different, such as i) drug export (the aberrant expression of ATP-binding cassette, ABC, drug pumps); ii) high survival (the inhibition of antiapoptotic processes, the high expression of proteins involved in DNA-damage repair, high telomerase activity); iii) reactive oxygen species (ROS) decrease (high ALDH activity, high expression of detoxification enzymes);and iv) the aberrant activation of pathways involved in stemness [15,16].

### 2.1. Drug Export in CSCs

It is common knowledge that the high expression of ABC proteins contributes to chemotherapy resistance and that CSCs overexpress different drug-transporter pumps, including ABCB1, ABCG2, and ABCC1 [15,17]. The Hoechst 33342 side population assay is a useful method to identify and isolate the CSC subpopulation in solid and hematopoietic tumors [18]. Yin et al. reported that CD133^+^ EPCAM^+^ liver CSCs express high levels of ABCG2 and ABCB1 and are highly resistant to doxorubicin treatment. The use of specific ABC inhibitors increases doxorubicin intracellular efflux, decreasing the sphere-forming capacity and viability of CSCs [19]. Other ABC transporters, including multidrug resistance protein 1 (MRP1, ABCC1), breast cancer resistance protein (BCRP), and MRP5/ABCC5, are reported as multidrug resistance transporters in solid and hematopoietic tumors [20,21,22]. Moreover, CD133^+^ melanoma CSCs expressed higher levels of ABCB5 compared to CD113^−^ cells and are resistant to the antiapoptotic activity of the natural compound caffeic acid phenethyl ester [23]. In the lung, the high expression levels of ABCB1 in CSCs mediated the resistance to PHA-665752 and crizotinib, a MET inhibitor [24]. Although the targeting of ABC transporters could be an effective strategy to target CSCs, the use of specific inhibitors causes many side effects, due to the expression of the same targets in normal cells, as well [24].

### 2.2. Enhanced Survival Ability in CSCs

CSCs can also circumvent the toxic effects induced by chemotherapeutic treatment activating DNA damage response (DDR) by the ATM(ataxia-telangiectasia-mutated)- and ATR (ATM- and RAD3-related)-dependent phosphorylation of targets such as Check-1, Check-2, or H2A.X (known as γH2A.X when phosphorylated) [25]. Manic and co-workers demonstrated that in colorectal CSCs the treatment with chemotherapeutic agents induces the activation of the DDR players, such as PARP1, RAD51, and/or MRE11, resulting in higher DNA damage repair machinery [26]. In breast cancer, both BRCA1^wt^ and BRCA1^mut^ CSCs were highly resistant to PARP inhibitors, due to the high expression of Rad51 and Sam68, and the inhibition of this critical signaling axis hampered CSC viability [27,28]. In addition, CD133^+^ glioma CSCs displayed resistance to radiotherapy treatment by the activation of DNA-damage repair mechanisms, where Check-1 and Check-2 are the main players. The inhibition of these two effectors reverted the radioresistance in glioma-CSCs, suggesting that targeting DNA damage could be a promising therapeutic approach for brain cancer treatment [29].

The deregulation of apoptotic pathways is another mechanism underlying CSC-mediated chemoresistance. A weak expression of death receptors, such as TRAIL and FAS, and the overexpression of inhibitor apoptosis proteins (IAPs) have been described in CSCs compared to differentiated tumor cells [6,30]. In CSCs, IAPs are often overexpressed and impair the activation of the apoptosis cascade by mediating pro-apoptotic protein degradation [31]. CD133^+^ colorectal CSCs highly resistant to 5-fluorouracil (5-FU) treatment expressed high levels of *SURVIVIN*, and the use of a specific aptamer-*SURVIVIN* siRNA enhanced the in vitro and in vivo 5-FU efficacy [32,33]. Moreover, in nasopharyngeal CSCs, XIAPs increased the stability of SOX2, and the use of an inhibitor of the IAP family in combination with 5-FU impaired tumor growth [34]. It has been reported that an aberrant expression of BCL-2 family members in CSCs contributes to drug resistance [20,35]. BCL-2 is overexpressed in leukemia stem cells and the use of a specific inhibitor of BCL-2, venetoclax, combined with azacitidine, resulted in disease remission in acute myeloid leukemia by CSC targeting [36,37]. In gastro-esophageal cancers, the use of the small molecule AT-101, which inhibits the BCL-2 family, decreased the expression of CSC markers (YAP1/SOX9) [38] (NCT00561197).

One of the effects of radiotherapy is the induction of DNA damage through the production of ROS and water-derived radicals. In CSCs, the presence of ROS is dramatically reduced due to the increase in ROS scavengers, limiting apoptosis induction and DDR mechanism activation [39]. The CSCs can also escape from anticancer therapies by increasing aldehyde dehydrogenase (ALDH) activity, which acts by reducing intracellular ROS levels. In turn, ROS generated from radio- and chemotherapy enhance the cytosolic expression of aldehydes, such as ALDH1A and 3A1 [40]. High levels of drug-metabolizing enzymes, such as ALDH1a1 and bleomycin hydrolase (BLMH1), have been characterized in the secretome of colorectal CSCs, increasing chemoresistance [41]. Moreover, several studies have pointed out that CSCs isolated from different tumor types display high ALDH expression levels and activity, which boost their chemoresistance [12,42]. Therefore, the upregulation of ROS levels could be an efficient strategy to counteract CSC features and sensitize CSCs to treatments.

### 2.3. Stemness Induction in CSCs by Different Signaling Pathways

Several signaling pathways, among which are Notch, Sonic-Hedgehog (SHH), Wnt/β-catenin, PI3K/Akt/mTOR (mTORC1 and mTORC2), TGF-β, JACK/STAT, and Hippo-YAP/TAZ, are aberrantly activated or deregulated in CSCs compared to normal stem cells [43].

The Wnt signaling pathway plays a key role during embryogenesis, and in many cancers, such as breast, colorectal, thyroid, and esophageal cancers, its activation promotes CSC growth and chemoresistance [43,44]. It has been demonstrated that the Wnt pathway is crucial for the maintenance of intestinal crypt homeostasis, and the APC mutation in transgenic mice increased the presence of LGR5+ stem cells at the bottom of crypts, boosting the transformation in microadenoma [45]. Vermeulen et al. demonstrated that the Wnt pathway is highly activated in CD133+ colorectal CSCs and can be influenced by extrinsic factors secreted by TME cells [46]. In hepatocellular carcinoma, the activation of the Wnt signaling pathway, induced by protein tyrosine kinase-2 (PTK2), boosted CSC tumorigenic potential and contributed to sorafenib resistance [47]. In endometrial CSCs, Lu and co-workers showed that SPARC-related modular calcium binding 2 (SMOC-2) interacts with Fzd6 and LRP6 (LDL-receptor-related protein 6) receptors and activates the Wnt/β-catenin pathway, increasing cisplatin and placlitaxel resistance [48].

The SHH signaling pathway is involved in normal embryogenesis development and plays a key role in the promotion of tumor growth and in drug resistance, upregulating the genes involved in CSC maintenance, such as *CD44*, *CCND2*, *c-MYC*, *NANOG*, *OCT4*, and *ALDH1* [49,50]. The SHH signaling pathway is involved in chemoresistance mechanisms by the modulation of the ABCG2 transporter and ALDH activity [51,52].

The dual TGF-β role in tumor progression has been extensively studied [53]. In fact, TGF-β is a key regulator of stemness, promoting EMT and radio-/chemoresistance [54,55]. Moreover, it has been shown that the cooperation of TGF-β with other signaling pathways increases CSC features. TGF-β and tumor necrosis factor alpha (TNF-α) induced a mesenchymal phenotype in breast CSCs by decreasing *CLDN3-4-7* gene expression and, in turn, increasing in vivo tumorigenesis and resistance to oxaliplatin, etoposide, and paclitaxel [56]. In leukemia stem cells, TGF-β regulated the activation of AKT and induced FOXO3a nuclear localization, boosting sphere-forming ability and tumor growth [57].

In addition to the mechanisms described above, other intrinsic and extrinsic factors contribute to drug resistance in CSCs. Increasing evidence sheds new light on the role of epigenetic alterations in increasing intratumoral heterogeneity and in the failure of standard therapies [58]. Several molecular mechanisms, such as DNA methylation, chromatin remodeling, regulation by non-coding RNAs, and the modification of histone proteins, contribute to the aberrant expression of ABC transporters in solid and hematological tumors [1,59]. Furthermore, numerous studies point out that the crosstalk between CSCs and the tumor microenvironment (TME) influences the plasticity of CSCs, promoting drug resistance [6,60].

To overcome this challenge in cancer treatment, many researchers have focused on the development of therapeutic approaches targeting CSCs and the different mechanisms involved in drug resistance. In this regard, NPs could be considered eligible candidates.

## 3. Natural Products as Adjuvant against Cancer Stem Cells

In recent years, due to rising drug costs and the need to protect the environment and give ecological credentials to chemotherapy compounds, the use of NPs is constantly growing and developing. In particular, NPs contain active compounds, which might affect multiple signaling pathways involved in self-renewal and the maintenance of CSCs, with limited side effects [61]. The use of these compounds in medicine has a historic background; in fact, the use of NPs has been reported since the time of the Egyptians for disease treatment. NPs can be extracted from several parts of plants, such as the root, stem, fruit, and leaf; dietary agents; and marine organisms [62,63].

The pharmaceutical industry shows great interest in NPs due to their unique properties, such as high diversity and steric complexity, lighter atoms, and low hydrophobicity [64]. These structural features can be used to synthesize commercial drugs useful for both cancer prevention and treatment [65]. Recent in vitro and in vivo studies have demonstrated that NPs counteract cancer progression by interfering with the self-renewal capacity of CSCs, the induction of apoptosis, the inhibition of cancer-cell spreading, and the arrest of the cell cycle [61]. Moreover, several natural compounds, such as alkaloids, terpenoids, polyphenols, and flavonoids are efficient modulators of ABC transporters and sensitize CSCs to conventional chemotherapeutic treatment [66].

To date, about 500 clinical trials are registered on the clinical trial.gov website, reporting the effects of NPs and NP-derived drugs for the treatment of different cancers (www.clinicaltrial.gov, accessed on 20 September 2022). Here, we report a large overview of the properties of NPs that sensitize CSCs to conventional chemotherapeutic treatments.

### 3.1. NPs Derived from Dietary Sources

A strong correlation has been demonstrated between tumor incidence and a correctly healthy lifestyle. Many NPs derive from food and belong to the polyphenols category. Polyphenols are characterized by the presence of aromatic benzene rings bonded to hydroxyl groups. These compounds are classified into stilbenes, lignans, tannins, flavonoids, and phenolic acids. Polyphenols show a role in the regulation of angiogenesis, inflammation, and the apoptosis of CSCs in in vitro settings [67]. Moreover, they can improve immune response by modulating T lymphocytes [68]. In this regard, recent studies have also highlighted that natural polyphenols can be used as adjuvants in association with conventional therapy to limit the CSCs’ drug-resistance phenomenon [69,70].

It has been demonstrated that curcumin, a polyphenol extracted from *Curcuma longa*, has anticancer effects against different types of tumors [71]. One of the prominent anticancer activities of curcumin is the blocking of NF-kB pathways through the inhibition of IKK activity [72]. Of note, liver cancer cells displayed different phenotypes after curcumin treatment, which can be divided into sensitivity and resistance. In sensitive cells, curcumin reduced cell viability via the reduction in CSC features, such as SP population, sphere-forming capacity, and tumorigenic potential. Conversely, curcumin treatment boosts stem-like properties in resistant cells. To identify the signaling pathways modulated by curcumin in sensitive and resistant cells, the authors performed a transcriptomic analysis which points out a downregulation of *HDACs* in sensitive cells. Curcumin, in combination with HDAC inhibitors, affected the sphere-forming ability and reduced the SP fraction in resistant cells [73]. In addition to the regulation of NF-kB pathways, curcumin modulates another key tumorigenic signal. Wu et al. reported that curcumin can inhibit the JAK2/STAT3 pathway in lung CSCs, reducing in vitro tumorsphere formation capacity and impairing tumor growth in a pre-clinical mouse model [74]. Moreover, this NP reduced the proliferation of and in turn promoted apoptosis in lung CSCs, causing a reduction in the main stemness markers through the downregulation of the Wnt/β-catenin and SHH pathways [75]. Curcumin, alone or in combination with piperine, inhibited the formation of tumorspheres in breast cancer cells, interfering with the stem-cell signaling pathways involved in carcinogenesis [76]. Moreover, curcumin could prevent the cell proliferation of LGR5^+^ colorectal cells by triggering autophagy and blocking via the TFAP2-mediated ECM pathway [77]. In recent years, several studies have shown that natural polyphenols may be used as adjuvant therapy in association with traditional treatment to reduce CSC drug resistance [69,70]. Curcumin is associated with low doses of cisplatin-induced apoptosis and reduced the migration in the CD166^+^/EpCAM^+^ CSC subpopulation in lung cancer cells by enhancing the sensitivity of the cells to chemotherapy [78]. In thyroid cancer, the combinatorial treatment with curcumin and cisplatin impaired sphere formation and the expression of stemness markers in thyrospheres via the downregulation of the JAK/STAT3 pathway [79]. Although different in vitro and in vivo studies have highlighted the role of curcumin in sensitizing CSCs to therapy, its use in clinical settings is limited by insolubility in water and fast metabolism [80].

Resveratrol is a natural product present in several types of food, such as the skin of grapes and berries, with multiple antitumoral effects, such as the inhibition of angiogenesis and detoxification enzymes and the induction of apoptosis [81,82]. In this regard, resveratrol can be considered a promising chemopreventive cancer agent. Jang et al. reported in a skin cancer mice model that the topical administration of resveratrol prevents tumor growth [83]. In osteosarcoma, resveratrol reduced cytokine synthesis (IL-6, TNF-α, IFN-γ, and oncostatin M) and inhibited STAT3 signaling to diminish the expression of CD133CSC markers [84]. According to Ferraresi et al., resveratrol could be used as a therapeutic strategy for the treatment of ovarian cancer, reducing cell migration and viability. Specifically, resveratrol counteracted the effect mediated by lysophosphatidic acid, inhibiting SHH signaling with the reduction in *BMI1*, a polycomb ring finger transcriptional factor involved in the activation of Hedgehog and restoring the autophagy pathway [85]. In pancreatic cancer, resveratrol impaired the stem-like features and the tumorigenic and invasive capacity of cancer cells [86]. In combination with 5-FU, resveratrol decreased the survival of CD133^+^ colorectal CSCs [87]. Another antitumoral resveratrol mechanism is the induction of oxidative stress. In breast cancer CSCs, resveratrol impaired mammosphere formation and xenograft tumor growth by inducing autophagy, with an increase in LC3-II, Beclin1, and Atg 7 expression levels, and reducing the Wnt pathway [88]. Moreover, this NP enhances the generation of ROS by overloading the mitochondrial electron transport chain, which ultimately influences cell apoptosis/necrosis and enhances cell death in colorectal cancer [89,90]. For these reasons, this compound can be effective in the inhibition of viability, tumorigenic potential, and self-renewal ability of CSCs. Of note, resveratrol modulates the crosstalk between TME and CSCs. In a multicellular TME system, resveratrol affected the interaction between colorectal CSCs and stromal cells and reduced the expression of stemness markers and sphere-forming capacity by blocking p65 NF-kB nuclear translocation [90]. The same mechanism of action has been reported in breast cancer. Resveratrol decreased the percentage of CD44^+^/CD24^−^ subpopulations and the expression levels of SOX2 and BMI-1 in BCSCs treated with the conditioned medium of cancer-associated fibroblasts [91].

Epigallocatechin-3-gallate (EGCG), a type of catechin found in green tea, has a chemopreventive activity against different types of cancers in vitro, in vivo, and in clinical settings [92,93]. Treatment with EGCG impaired the in vivo growth of prostate, lung, and gastrointestinal cancer cells [94]. In particular, EGCG regulated the expression of CSC markers and, thus, CSC features [95,96]. Luo et al. hypothesized the use of EGCG in colon cancer prevention and treatment as dietary supplements or adjuvant therapy, due to EGCG’s anti-proliferation and anti-migration properties [97]. EGCG treatment reduced the invasive capacity and induced apoptosis through the downregulation of the STAT3 pathway and the modulation of proteins involved in EMT and apoptosis, impaired Wnt pathway activation, and increased the sensitivity to 5-FU treatment in colorectal CSCs [97,98,99]. In lung cancer, EGCG targeted CD133^+^ cells, decreasing the self-renewal and tumorigenic potential of CSCs through the regulation of the circadian rhythm protein CLOCK [100]. Moreover, EGCG downregulated the Wnt pathway and reduced the proliferation and stemness marker expression in lung CSCs [101].

Flavonoids are polyphenolic compounds found in nuts, teas, fruit, and vegetables with antioxidant, anti-inflammatory, and anticancer properties [102]. The classification of flavonoids depends on their level of oxidation and includes flavanones, flavones, flavanols, and anthocyanins [103]. Several studies hypothesize that diets containing a high number of flavonoids could have cancer chemopreventive effects, targeting CSCs [104].

Citrus fruits including *Citrus depressa* (shiikuwasa), and *Citrus sinensis* (oranges) contain nobiletin, a healthy dietary polymethoxylated flavone [105] with a variety of biological actions, including anti-inflammatory, anti-tumor, and neuroprotective effects [106,107]. Nobiletin enhanced the internalization of chemotherapeutic or other natural compounds viathe inhibition of ABC transporters [108,109]. In non-small cell lung cancer, treatment with nobiletin inhibited the Wnt pathway and negatively correlated with EMT and stemness, reducing CD133 and ALDH1 stem markers [110]. Our group recently demonstrated that nobiletin and xanthohumol—a prenylated flavonoid contained in hop cones—extracts reduced the viability of colorectal CSCs and synergized with FOX (5-FU plus oxaliplatin) in inducing apoptosis and reducing stemness features, such as CD44v6 expression and Wnt pathway activation [111].

Apigenin is a bioavailable flavonoid belonging to the flavone class and is present in vegetables, fruits, and drinks. Apigenin exhibits anti-inflammatory activities, antioxidant effects, and anticancer properties [112,113]. Erdogan et al. demonstrated that adjuvant therapy with apigenin enhanced the sensibility of prostate CSCs to cisplatin treatment. This combinatorial therapy increased the cisplatin-induced apoptosis via the downregulation of *BCL-2*, *SHARPIN*, and *SURVIVIN* mRNAs, and enhanced the expression levels of caspase-8, Apf-1, and P53 [114]. In breast cancer, apigenin reduced the CD44+/CD24− subpopulation in triple-negative breast CSCs, inducing tumor shrinkage through the downregulation of YAP/TAZ activity [115]. Moreover, apigenin in combination with cisplatin reduced the tumorigenic potential of CD133^+^ lung CSCs [116].

Quercetin is a secondary metabolite from fruit and vegetable flavonols with anti-inflammatory and antioxidant properties. Cao et al. described howquercetin-3-methyl ether impaired the sphere-forming capacity of breast CSCs by reducing the expression of stemness genes (*SOX2, NANOG*) and the Notch and PI3K/AKT pathways [117]. Moreover, quercetin inhibited the tumor growth and metastasis formation of CD44^+^/CD24^−^ CSCs and reduced the expression levels of ALDH1A and CXCR4 [118,119]. In CD24^+^/CD133^+^ pancreatic CSCs, quercetin reduced the activation of the Wnt pathway and the expression of stemness markers [120].

Naringin and naringenin are flavanones obtained from citrus fruits with anti-inflammatory, antioxidant, and antitumoral properties. These natural products display chemopreventive activity in many tumors, such as lung and colorectal cancers [121,122]. Many studies have pointed out that naringin and naringenin hamper tumor growth and progression by inhibiting pathways involved in survival, apoptosis, ROX detoxification, autophagy, and metastasis formation [123]. By bioinformatics analyses, Hermawan et al. identified naringenin as a potential drug to target breast CSCs. Indeed, naringenin treatment decreased the sphere-forming and colony-forming capacity, migration, and expression of stemness-related genes (*CTNNB1*, *ALDH1*, *VIMENTIN*) in breast CSCs [124]. In cervical cancer spheroids, naringenin in combination with cisplatin reduced the cell viability and invasion of cancer cells [125].

Sulforaphane (SFN) is an active isothiocyanate derived from the hydrolyzation of glucoraphanin by myrosinase activity. SFN is a phytoconstituent that belongs to the *Brassicaceae* family, which includes vegetables such as cauliflower, kale, cabbage, and broccoli. Numerous manuscripts have highlighted SFN’s anticancer effects in both in vitro and in vivo models in different tumors, acting as an epigenetic modulator that induces apoptosis and senescence [126,127]. Li et al. highlighted that SFN treatment decreased the percentage of CD133+and ALDH+ cells in lung spheroids induced after exposure to cisplatin and enhanced the in vivo antitumor effect of cisplatin [128]. In breast cancer, SFN impaired the formation of mammospheres via the reduction in ALDH+ cells and the activation of the Wnt/β-catenin signaling pathway. Moreover, SFN treatment decreased tumor growth and the second engraftment of breast CSCs [129]. Castro et al. showed that SFN can modulate the cell proliferation, tumorsphere formation, cell viability, and phenotype of CSCs derived from triple-negative breast cancer and counteract the xenograft tumor growth in mice [130].

Fisetin is another flavonol found in some vegetables and fruits including onion, cucumber, apple, grape, and strawberry. Fisetin is a neuroprotective agent and acts as a chemopreventive/chemotherapeutic agent in different cancers [131]. In lung CSCs, treatment with fisetin inhibited cell growth by modulating mTOR and PI3K/AKT signaling and decreased the number of colonies in a dose-dependent manner [132]. Another study in lung cancer demonstrated that fisetin exhibits anti-invasion and anti-proliferative effects via the downregulation of CD44 and CD133 stem-like markers [133]. In Table 1, we summarize the studies previously described regarding the effect of NPs on CSCs (Table 1).

### 3.2. NPs Derived from Botanical Sources

Different drugs, derived from natural compounds, are used in clinical practice as anticancer agents. Paclitaxel isolated from *Taxus brevifolia* was one the first anticancer agents studied in ovarian and breast adenocarcinoma. Some plants can produce toxic substances such as phenol or tannin when attacked by predators [134]. In this regard, many researchers studied plants as a possible source of NPs to counteract CSCs.

Luteolin is a flavone present in about 300 plant species [135]. Luteolin impaired the expression of the stemness markers ABCG2 and CD44 and affected the ALDH1 activity and spheroid formation capacity of breast CSCs. Moreover, luteolin sensitized CSCs to taxol treatment [136]. Luteolin in combination with quercetin impaired the sphere-forming ability and the expression levels of NANOG, SOX2, and CD44 [137]. In oral CSCs, luteolin effectively reduced proliferation, ALDH activity, and CD44, inactivating the IL6/STAT3 axis [138].

Berberine (BBR) is an isoquinoline alkaloid obtained from the roots and stems of anti-inflammatory plants and can induce apoptosis by reactive oxygen generation [139,140]. In the literature, the ability of BBR as a modulator of epigenetic modification has been widely demonstrated [141,142]. Zhao et al. showed that BBR impairs the proliferation and sphere-forming capacity of colorectal CSCs by enhancing the expression levels of p27 and p21, increasing the percentage of cells in the G1/G0 phase and, in turn, reducing CD44 and CD133 markers. In line with its epigenetic modulator activity, BBR impaired the RNA m6A methylation levels. Moreover, BBR treatment affects the tumorigenic capacity of colorectal CSCs and sensitizes the cells to 5-FU and irinotecan [143]. In pancreatic cancer, BBR and gemcitabine combinatorial treatment decreased SP percentage and the expression of the stemness genes *POU5F1*, *SOX2*, and *NANOG* [144]. Moreover, BBR can counteract sphere formation, the expression of EMT stemness markers, and the activation of the *GLI1–BMI1* axis induced by chemotherapy in ovarian CSCs [145]. In neuroblastoma, BBR treatment was able to induce the expression of epithelial-like marker E-cadherin, downregulating crucial signaling pathways that regulate tumor progression, such as PI3K/Akt, TGF-β, and MAPK [146].

Vincristine is another alkaloid, derived from the Madagascar periwinkle, *Catharanthus roseus*, with anti-tumor activity. Its biological mechanism of action is based on the inhibition of microtubule aggregation and, consequently, the arrest of cell mitosis in metaphase [147]. The treatment of a neuroblastoma cell line (SH-SY5Y) with vincristine reduced cell proliferation in a dose-dependent manner by blocking the cell cycle in the G2-M phase with increased cyclin B expression and decreased cyclin D levels. Overall, these data showed that vincristine could be a promising chemotherapeutic agent for the treatment of neuroblastoma [148]. For decades, vincristine has been used in combination with chemotherapy in different tumors, as well as acute myeloid leukemia [149], lung cancer [150] (NCT00003847), colorectal cancer [151], and breast cancer [152]. Nevertheless, its anti-proliferative effects in CSCs have been little shown. Moon et al. highlighted that vincristine influences the methylation state of the runt-related transcription factor-3 gene (*RUNX3*) in colorectal cancer. The treatment with this alkaloid induced the demethylation of *RUNX3*, leading to the recovery of *RUNX3* mRNA expression in colorectal cancer cells without affecting DNA methylation in healthy colon cells [153]. These observations suggest a potential therapeutic approach for CSC targeting.

Alkaloids gained from the bark of the *Cinchona officinalis* tree have been used for more than a century for malaria prevention and treatment. Among these alkaloids, chloroquine (CQ), derived from quinine, is a potent inhibitor of autophagy in cancer cells [154]. Cufì et al. reported that CQ treatment slightly reduced the CD44^+^/CD24^−^ stem-like subpopulation and vimentin expression in triple-negative breast cancer (TNBC) cells [155]. Moreover, CQ in combination with paclitaxel decreased ALDH-positive and CD44^+^/CD24^−^ cell subpopulations and the sphere-forming capacity of TNBC cells. The combination treatment impaired autophagy through the upregulation of LCB3-II and p62 expression levels and enhanced apoptosis and cleaved caspase-3 levels by inhibiting the JAK/STAT pathway. In in vivo settings, CQ and paclitaxel lessened in vivo tumor growth and lung metastatic foci [156]. Liang et al. showed that CQ targets breast CSCs by inducing mitochondrial depolarization and the accumulation of DNA double-strand breaks, and CQ in combination with carboplatin diminished autophagy and the expression levels of proteins involved in DNA repair machinery [157]. CQ affected the CD133^+^ subpopulation and the tumorigenic potential of pancreatic CSCs and patient-derived xenografts (PDXs) through a mechanism not previously described. In fact, CQ inhibited the CXCL12/CXCR4 axis and SHH pathway in pancreatic- CSCs [158]. A similar effect of CQ treatment was also observed in esophageal squamous cell carcinoma CSCs [159].

Recent studies have revealed that capsaicin, derived from plants of the genus *Capsicum*, showed considerable anticancer effects [160]. Zhu et al. reported that capsaicin decreases sphere size, hampers CSC survival in a dose-dependent manner, and downregulates markers such as CD133, CD44, OCT-4, NANOG, and SOX2, typically expressed in prostate CSCs. Moreover, the authors showed that capsaicin interfered with the Wnt/β-catenin signaling pathway in prostate CSCs. Briefly, capsaicin drastically reduced GSK3 phosphorylation and avoided β-catenin’s translocation into the nucleus, downregulating target genes such as *MYC* and *CCND1*. The activation of the Wnt/β-catenin pathway restored the sphere-forming ability of prostate CSCs and induced the upregulation of the above-described stemness markers [161].

A saturated derivative of capsaicin, dihydrocapsaicin (DHC), is a potent inducer of autophagy [162]. DHC-induced autophagy in a catalase-dependent manner in colon and breast cancer cell lines has been reported. Oh and co-workers showed that DHC induced breast and colorectal cancer cell arrest in G0-G1, upregulating the expression levels of autophagy-related proteins [163]. Thanks to its ability to induce autophagy, DHC could be considered a promising anticancer agent. In this regard, DHC efficiently targeted the CD133^+^ neural cell population, inducing cell death [164] (US20090076019A1). However, due to its low bioavailability, DHC isno longer being tested as a CSC-targeting agent (Table 2).

### 3.3. NPs Derived from Marine Sources

The marine microenvironment is a heterogeneous environment, characterized by unique conditions (low oxygen and sunlight, as well as high salinity and pressure) that favor the presence of micro- and macro-organisms producing particular metabolites. It has been demonstrated that these molecules with unique biochemistry structures, containing various heterocyclic rings and diverse heteroatoms, can be used to prevent and treat cancer [62]. Given the growing number of marine natural compounds (MNC) used in medicine, more researchers have focused on the structure and synthesis of analogs with anticancer properties [165].

Nortopsentin, a bis-indolyl alkaloid isolated from deep-sea sponges (*Spongosoritesruetzleri*), exhibits significant antitumor activity against P388 murine leukemia [166]. Cascioferro et al. synthesized nortopsentin analogs by introducing the substitution of a central imidazole ring with a 1,2,4-oxadiazole motif and a 7-azaindole in place of the original indole motif. Among these compounds, 1k and 1n displayed cytotoxic effects on MCF-7, Caco-2, HeLa, and HCT-116 cells. The anti-proliferative activity on MCF-7 of these compounds was associated with a pro-apoptotic activity involving chromatin condensation and membrane blebbing. Moreover, these chemicals induced a buildup of cells in the G0-G1 phase, indicating that they could influence DNA replication [167]. Similar results were obtained by Di Franco et al. using another analogous of nortopsentin, NORA234. This compound reduced, at early timepoints, the clonogenic potential and the proliferation rate of colorectal CSCs. However, NORA234’sprolonged administration drove the selection of resistant subclones, characterized by high expression levels of CD44v6 and β-catenin activity, with an increased CHK1-driven DNA damage response. Treatment with NORA234 in combination with CHK1 inhibitor enhanced apoptosis and hampered the proliferation and clonogenic capacity of colorectal CSCs together with the decrease in CD44v6^+^/Wnt^high^ subpopulations [168].

Renieramycin M (RM) is the major bis-tetrahydroisoquinolinequinone alkaloid derived from the blue sponge *Xestospongia* species. Treatment with non-toxic concentrations of RM reduced colony and spheroid formation and the expression of CD133, CD44, and ALDH1A1 stem-like markers in lung CSCs [169]. According to these findings, RM could be considered a promising anticancer compound.

Fucoxanthin and its metabolite fucoxanthinol (FxOH), carotenoids isolated from different brown algae species, exhibit beneficial cancer prevention and anticancer features [170]. Terasaki et al. demonstrated that FxOH impaired the growth, sphere-forming ability, and tumorigenic potential ofCD44^high^/EpCAM^high^ colorectal CSCs by the inactivation of AKT signaling and the downregulation of PPARβ/δ and PPARγ protein expression levels [171]. Moreover, FxOH treatment reduced the expression levels of N-cadherin and vimentin EMT markers, which correlate with lessened levels of glycine and succinic acid, in colorectal CSCs [172]. Sulfated polysaccharides called fucoidans derived from brown algae have a variety of biological functions. According to Vishchuk and colleagues, sulfated (1→3)-L-fucan, obtained by *Saccharinacichorioides*, reduced the colony-formation capacity of different cancer cell lines [173].

Bryostatin-1 is a macrocyclic lactone of marine origin derived from the *Bugula neritina* invertebrate. Different pre-clinical and clinical studies demonstrated its role as an antitumor agent [174,175]. Sikorska et al. showed that among different natural compounds, bryostatin-1 promoted a differentiated state in melanoma CSCs, reducing their high proliferative rate and the ABCB5^+^ subpopulation [176]. Bryostatin-1 increased the Gleevec-mediatedapoptosis of chronic myeloid leukemia SCs, reducing the fraction of G0/G1 CD34^+^ cells [177] (Table 3) (Figure 1).

### 3.4. Other Natural Compounds

Other NPs such as retinoids, which do not fit into the classifications of polyphenols, flavonoids, or alkaloids, have shown promising effects for targeting CSCs [178]. Retinoids, which are classified as terpenes, induce the differentiation of CSCs, making them more sensitive to chemotherapeutic agents [179]. All-Trans Retinoic Acid (ATRA) is a biologically active compound belonging to the retinoid group and is a metabolite of vitamin A. It is essential for a variety of biological processes, such as cell division, organogenesis, differentiation, and cell death. ATRA can inhibit ALDH activity and revert MDR in CSCs. Ginestier et al. displayed that ATRA decreased mammosphere formation by regulating signaling pathways involved in CSC differentiation [180]. In glioblastoma, treatment with ATRA increased the expression levels of astrocytic (GFAP) and neuronal (TUJI) markers and reduced proliferation and self-renewal in neurospheres through the modulation of the ERK1/2 pathway [181]. Furthermore, ATRA impaired the in vitro and in vivo proliferation of CSCs by decreasing the expression of OCT4, SOX2, Nestin, and CD44 and the activation of Wnt/β-catenin signaling in head and neck cancer [182]. In lung cancer, ATRA, in combination with gefitinib, reduced the ALDH1A1^high^/CD44^high^ CSC subpopulation and growth boosted by chemotherapy [183]. Several studies have shown how ATRA’s epigenetic processes work [184,185].

Although the use of NPs might be faced with some problems, such as screening difficulties, NPs are characterized by peculiar biological structures, which make them an appealing approach to anticancer therapy [186].

## 4. Natural Products in Clinical Trial for Cancer Treatment

The high costs and side effects of chemotherapy and radiotherapy have led to a great interest in natural medicine. NPs being readily available and more tolerable in comparison with synthetic compounds make them attractive agents for cancer treatment [187]. Many NPs, such as curcumin, EGCG, resveratrol, quercetin, and apigenin, have shown strong anticancer effects in numerous pre-clinical studies, but the feasibility of translating the efficacy of these compounds in clinical trials is an ongoing challenge. Differences in genetics and metabolism between pre-clinical models and humans, as well as the solubility and the time of action of these compounds, could limit their use in clinical settings. Pharmacokinetics and pharmacodynamics studies could better elucidate NPs’ effects on humans. Despite several pre-clinical studies showing the therapeutic potential of curcumin, a few clinical trials assessed the effectiveness of curcumin in the treatment of cancer patients due to its reduced bioavailability [188]. Nonetheless, the use of curcumin as a substitute for corticosteroids (standard therapeutic agents) in combination with immunomodulatory compounds (lenalidomide) or as a proteasome inhibitor (bortezomib) in 15 multiple myeloma patients showed progression-free survival without the negative side effects linked to the steroid-based combination therapy [189]. Moreover, in multiple myeloma and prostate cancer patients, curcumin and piperine in combination delayed cancer progression (NCT04731844). The effect of curcumin on myeloma patients was also evaluated in a pilot randomized clinical trial. The treatment with melphalan, prednisone, and curcumin displayed an increased overall remission with a reduction in IL-6, VEGF, and TNF-α levels compared with myeloma patients treated with melphalan and prednisone alone [190] (ISRCTN14131419). In another clinical study enrolling 150 patients with advanced or metastatic breast cancer, the intravenous administration of curcumin (300mg) with paclitaxel (80 mg/m^2^) for 12 weeks induced a tumor reduction of 50.7% [191] (NCT03072992). There are many active clinical trials to evaluate curcumin’s chemopreventive, neoadjuvant, and radioprotective effects in breast cancer patients (NCT01975363, NCT03847623, NCT01246973). The phase IIa CUFOX clinical trial assessed the safety and effect of curcumin in combination with FOLFOX-based chemotherapy in metastatic colorectal cancer patients [192] (NCT01490996).

EGCG is a polyphenol with multiple antitumor activities [93]. In a second phase clinical trial, the effects of indole-3-carbinol (I3C) and EGCG in combination with taxane and platinum-based chemotherapy were tested in stage III–IV serous ovarian cancer patients. Patients treated with I3C and EGCG plus chemotherapy showed an increased median overall survival (60 months) and median progression-free survival (48.5 months) compared with single treatment and chemotherapy alone. Moreover, I3C and EGCG treatment reduced cancer recurrence [193] (ACTRN12616000394448). In a phase II study, EGCG treatment was evaluated in chemotherapy-treated advanced lung cancer patients who had developed acute radiation esophagitis as a side effect. Both EGCG preventive treatment and EGCG administration in radiation-treated patients decreased the esophagitis grade and the serum levels of pro-inflammatory factors compared with standard treatments in lung cancer patients [194] (NCT02577393). Similarly, a phase II study evaluated the effect of EGCG in esophageal cancer patients with esophageal obstruction [195] (NCT05039983). In bladder cancer patients, a phase II randomized pre-clinical trial assessed the effect of Polyphenon E (a green tea polyphenol formulation in which EGCG is a main component) in neoadjuvant therapy before the transurethral resection of a bladder tumor or cystectomy. Although there are no differences in EGCG tumor levels between the EGCG-treated and placebo patient groups, a dose-dependent downregulation of two tumor biomarkers, clusterin (apoptosis marker) and PCNA (proliferation marker), was observed in EGCG-treated patients, supporting the chemoprotective activity of this compound [196]. The impact of Polyphenon E on serum and tissue levels of progression biomarkers was characterized in a randomized phase II single-arm open-label clinical study including breast cancer patients. In particular, Polyphenon E neoadjuvant daily administration induced a decreasein serum HGF levels without altering those of VEGF (NCT00676793). The serum HGF and VEGF levels together with the measurement of oxidative damage and inflammatory biomarkers were also evaluated in a phase IB randomized dose-escalation trial in stage I–III hormone receptor-negative breast cancer patients treated with Polyphenon E for 6 months. After the treatment with adjuvant therapy, a significant, but transient, decrease in serum HGF and VEGF was observed [197].

Although many in vitro and in vivo studies have highlighted the potential use of resveratrol in clinical settings, a limited number of clinical trials have been carried out [198]. A phase I clinical study in breast cancer found that resveratrol was tolerated throughout a 12-week treatment period on 39 patients and increased levels of resveratrol were found in blood serum patient samples. After resveratrol treatment, no significant changes in p16, CCND2, APC, orRASSF-1α DNA methylation were observed, but only a decreased methylation profile of RASSF-1α and an increase in the APC profile [199]. These findings imply that resveratrol may operate as a chemopreventive agent for breast cancer by affecting the epigenetics of breast cancer-related genes. Alternatively, in a phase I clinical study, treatment with MPX (pulverized muscadine grape skin composed ofellagic acid, quercetin, and resveratrol) in biochemically recurrent prostate cancer patients (BRPC), at different concentrations (500 mg up to 4000 mg/day), was shown to be safe and tolerable [200]. Taken together, these data result in the possibility of investigating MPX’s effects in a randomized, multicenter phase II trial. In 112 patients of BRPC enrolled in a randomized, multicenter, placebo-controlled clinical trial, no significant difference was observed in terms of PSA doubling time in control and MPX-treated cohorts. Moreover, within the clinical trial, the authors identified a patient population that could benefit from treatment with MPX, but further studies are needed [201].

In another phase I pilot trial (NCT00256334), the effects of low doses of resveratrol derived from plants and resveratrol-containing freeze-dried grape powder (GP) were evaluated on Wnt-signaling modulation in colon cancer patients. Resveratrol/GP treatment (80 g/day containing 0.07 mg of resveratrol) significantly inhibited Wnt target gene expression (myc, jun, TCF7, cyclin D1, axin II) in healthy colonic mucosa without effects on tumor mucosa. This study highlights that GP treatment could have a role in the prevention of colon tumor formation [202]. In this clinical pilot study, SRT501, a micronized form of resveratrol, was administered to patients with colorectal cancer and hepatic metastases, who were scheduled to undergo hepatectomy, at a dose of 5.0 g daily for 14 days. This treatment method increased drug availability and absorption. After 1–2 weeks of therapy with resveratrol or SRT501, the observed amounts of parent resveratrol and its primary metabolites in the colon tissue of patients were comparable to the efficacious doses of resveratrol utilized in pre-clinical investigations. In addition, cleaved caspase-3, an indication of death, dramatically increased in malignant hepatic tissue after SRT501 therapy by 39% in comparison to tissue from patients who received a placebo [203]. Furthermore, a similar first-phase clinical trial has also confirmed the diminishing of ki-67 levels (a proliferation marker) in colorectal cancer patients [204] (NCT00433576). While overall data suggest that resveratrol has some pharmacological properties, it is uncertain if these effects are sufficient to make it an effective treatment agent for colon cancer. To date, 11 of 16 marine drugs are used in the treatment of different cancers [205]. In particular, Plitidepsin (Aplidin^®^, produced by PharmaMar) is a drug approved in Australia for multiple myeloma leukemia and lymphoma [206]. Polatuzumabvedotin (peptide derived from marine cyanobacteria), by inhibiting tubulin polymerization, induced CSC death. This NP was approved in 2019 by the FDA for the treatment of non-Hodgkin lymphomas, chronic lymphocytic leukemia, and B-cell lymphomas [207]. Lurbinectedin (a synthetic derivative of trabectedin) showed anticancer activity by the degradation and inhibition of RNA polymerase II; in 2020, it was approved for metastatic small lung cancer treatment [208]. In this regard, other molecules derived from the marine environment are undergoing clinical evaluation.

Vincristine belongs to antimitotic agent groups that interfere with microtubule organization. Several pre-clinical studies have demonstrated its role as an anticancer agent. Nonetheless, it has been shown that vincristine provokes neurotoxicity, suggesting its use at low dosages. In several clinical trials, vincristine was used at low concentrations in combination with doxorubicin, dacarbazine, methotrexate, and also rituximab [209]. In a clinical study carried out on children affected by low-grade glioma, treatment with vincristine and carboplatin was defined as eligible first-line therapy, representing the first European clinical randomized study and the second of European chemotherapy in childhood LGG [210] (European Union Clinical Trials Register No. 2005-005377-29). In an open-label, multicentre II phase clinical trial, pretreatment with ofatumumab (antibody against CD20) and miniCHOP (a combination with vinacristine, reduced-dose cyclophosphamide, prednisone, and doxorubicin) in 80-year-old patients improved overall survival in comparison with standard therapy [211] (NCT01195714).

Bryostatin-1 showed different effects mediated by the modulation of protein kinase PKC activity. Due to a lack of pharmacokinetics data in humans, the first clinical trials were hampered [212]. Alone or in combination with another drug, bryostatin-1 has been evaluated in phase I and II clinical trials. These multiple trials showed that this lactone, alone or in combination with other compounds, exerts synergistic anti-tumor activity. In a phase II trial, patients with metastatic renal carcinoma, treated with an intravenous infusion of bryostatin-1 with formulation PET (polyethyleneglycol, ethanol, and Tween 80), responded to the treatment without severe adverse effects [213]. The treatment of 25 patients with chronic lymphocytic leukemia (CLL) and relapsed low-grade non-Hodgkin lymphoma with bryostatin-1 resulted in one patient in complete remission and two in partial remission. Moreover, this treatment promoted a differentiative state of CLL cells, demonstrated by the presence of CD11c/CD22/CD20 B-cell subpopulation [214]. Nevertheless, Bryostatin-1 is not very available in nature, and it could need to be synthesized.

Chemicals produced from cruciferous vegetables, such as sulforaphane (SFN), a breakdown product of glucoraphanin, may help inhibit prostate cancer development and progression. A double-blind, randomized controlled trial, conducted on ninety-eight men scheduled for prostate biopsy, evaluated the effect of broccoli sprout extract (BSE) on the expression of different prostate cancer biomarkers such as histone H3 lysine 18 acetylation (H3K18ac), HDAC3, HDAC6, Ki67, p21, and histone deacetylase (HDAC). Unfortunately, BSE-treated patients did not significantly display a reduction in HDAC activity or prostate tissue biomarkers. By performing an RNA-seq analysis on prostate biopsies, 40 differently expressed genes linked to BSE treatment were characterized, including two prostate cancer-related genes, AMACR and ARLNC1. According to this study, supplementing with BSE is associated with alterations in gene expression but not with changes in prostate tissue biomarkers [215] (NCT01265953). Furthermore, an interventional clinical trial evaluated the effect of a diet rich in broccoli in reducing the risk of cancer progression, especially in men diagnosed with low- and intermediate-risk prostate cancer on active surveillance. Trans-perineal template biopsies from forty-nine men on active surveillance, who were fed different glucoraphanin-rich broccoli soups for 12 months, were analyzed by RNA sequencing. The obtained results displayed a reduced expression of genes linked to inflammation processes and EMT in men consuming the glucoraphanin-rich broccoli diet. Although the trial lacked the necessary power to evaluate clinical progression, an inverse relationship was found between the consumption of cruciferous vegetables and a decreased risk of prostate cancer advancement [216] (NCT01950143).

In a phase I clinical study, 37 colorectal cancer patients treated with chemotherapy were randomized to receive either 100 mg fisetin (n = 18) or placebo (n = 19) for seven consecutive weeks. Fisetin administration reduced the plasma levels of IL-8, hs-CRP, and the expression of MMP-7. Accordingly, fisetin might decrease the inflammatory state in colorectal cancer patients, supporting fisetin treatment as a potential supplementary anticancer drug for these patients and warranting future investigations [217] (code: IRCT2015110511288N9) (Table 4).

## 5. Conclusions and Perspectives

In recent years, the rising costs of cancer treatment and the urgent need for eco-sustainability endorse a new paradigm in oncology, known by the term “Green Oncology”. To date, the ecological model, in which oncologists consider not only individual illness but also population health as a component of the biosphere, is increasingly replacing the biomedical and biopsychosocial ones. In this context, Green Oncology’s aim is to preserve the environment and the ecosystem by promoting the use of NP-derived drugs, which avoids treatments with chemotherapeutics that are not easily disposable and are also characterized by fewer side effects. Compelling evidence points out that NPs effectively lessen the stem-like properties of CSCs, which are refractory to standard and targeted therapies. In this review, we reported the most appealing pre-clinical studies regarding the ability of NPs to lessen the expression of CSC markers and the activation of pro-tumorigenic signaling pathways. Despite the promising results obtained with the use of NPs in in vitro systems and pre-clinical models, the limited systemic availability of these all-natural molecules as well as their faster metabolism pose a challenge to their efficacy in targeting cancer cells in organs far from the site of absorption. Although NPs could really improve the malignant progression of tumors, further efforts are needed to reduce the timeline of bench to bedside.

## Figures and Tables

**Figure 1 jcm-11-06996-f001:**
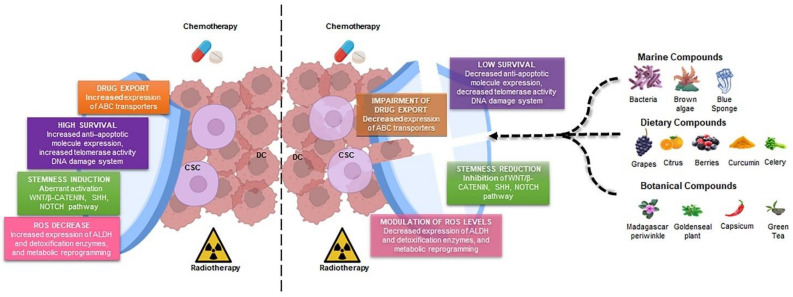
Natural products destroy the shield of CSCs. Radio- and chemotherapy target differentiated cancer cells, which represent most cells within the tumor mass, sparing the CSC subpopulation, characterized by the high expression of ABC transporters, anti-apoptotic molecules, and detoxification enzymes and the aberrant activation of stemness pathways and DNA repair machinery (**left part**). Natural products derived from different sources (marine, food, and botanical compounds) are able to reduce CSC features and increase their sensitivity to radio- and chemotherapy (**right part**). DCs, differentiated cells; CSC, cancer stem cell; ABC, ATP-binding cassette.

**Table 1 jcm-11-06996-t001:** NPs derived from dietary sources and effects on CSCs.

Natural Products (Source)	Tumor Type	Effects on CSCs	Alone or in Combination with Other Compounds	References
Curcumin (*Curcumalonga*)	Liver cancer	In vitro reduction in SP subpopulation and sphere formation.	HDAC inhibitors	[73]
	Lung cancer	In vitro reduction in tumorsphere formation and inhibition of JAK2/STAT3 pathway. In vivo impaired tumor growth.		[74]
	Lung cancer	In vitro arrest of cell proliferation, induction of apoptosis, and reduction in the main stemness markers via Wnt/β-catenin and SHH pathways downregulation.		[75]
	Breast cancer	In vitro reduction in tumorsphere formation	Piperidine	[76]
	Colorectal cancer	In vitro reduction in LGR5+ cell proliferation and induction of autophagy.		[77]
	Lung cancer	In vitro induction of apoptosis, reduction in CD166^+^/EpCAM^+^ cell migration.	Cisplatin	[78]
	Thyroid cancer	In vitro impairment of sphere formation and reduction in stemness marker expression via JAK/STAT3 downregulation.	Cisplatin	[79]
Resveratrol (skin of grapes and berries)	Osteosarcoma	In vitro inhibition of STAT3 pathway and reduction in CD133 expression.		[84]
	Ovarian cancer	In vitro reduction in migration and viability, inhibition of SHH pathway, and induction of autophagy.	Lysophosphatidic acid	[85]
	Pancreatic cancer	In vitro reduction in stem-like features. In vivo reduction in tumorigenic and invasive potential.		[86]
	Colorectal cancer	In vitro reduction in CD133^+^ cell survival.	5-FU	[87]
	Breast cancer	In vitro reduction in tumorsphere formation via autophagy induction and Wnt pathway reduction. In vivo reduction in tumor growth.		[88]
	Colorectal cancer	In vitro reduction in stemness markers and sphere formation.		[89,90]
	Breast cancer	In vitro reduction in CD44^+^/CD24^−^ subpopulations and SOX2 and BMI1 expression levels.		[91]
EGCG (green tea)	Colorectal cancer	In vitro reduction in invasive capacity and induction of apoptosis and chemosensitivity via STAT3 and Wnt pathway downregulation.		[99]
	Lung cancer	In vitro reduction in self-renewal of CD133^+^ cells. In vivoreduction in tumorigenic potential.		[100]
	Lung cancer	In vitro downregulation of Wnt pathway and reduction in proliferation and stemness marker expression.		[101]
Nobiletin (*Citrus depressa* and *Citrus sinensis*)	Lung cancer	In vitro reduction in Wnt pathway and CD133 and ALDH1 expression levels.		[110]
	Colorectal cancer	In vitro reduction in cell viability, CD44v6 expression, Wnt activation, and induction of apoptosis.	Xanthohumol (flavonoid) and FOX	[111]
Apigenin (fruits, vegetables, and drinks)	Prostate cancer	In vitroinduction of apoptosis and inhibition of cell migration and NF-kB pathway.	Cisplatin	[114]
	Breast cancer	In vivo reduction in CD44^+^/CD24^−^ tumorigenic potential via downregulation of YAP/TAZ pathway.		[115]
	Lung cancer	In vivoreduction in CD133^+^ tumorigenic potential.	Cisplatin	[116]
Quercetin (fruits and vegetables)	Breast cancer	In vitro reduction in tumorsphere formation and decrease in stemness gene (*SOX2, NANOG*) expression and Notch and PI3K/AKT pathway activation.		[117]
	Breast cancer	In vitro reduction in ALDH1A1 and CXCR4 expression levels. In vivo reduction in CD44^+^/CD24^−^ tumor growth and metastasis formation.		[118,119]
	Pancreatic cancer	In vitro reduction in Wnt pathway activation and stemness markers.		[120]
Naringine and naringenin (*Citrus fruits*)	Breast cancer	In vitro reduction in mammosphere and colony formation and migration, decrease in stem-like markers (β-catenin, ALDH1).		[124]
	Cervical cancer	In vitro reduction in cell viability and invasive capacity.	Cisplatin	[125]
Sulforaphane (*Brassicaceae*)	Lung cancer	In vitro decrease in CD133+ and ALDH+ cells.	Cisplatin	[128]
	Breast cancer	In vitro reduction in mammosphere formation and decrease in ALDH expression and Wnt/β-catenin pathway activation. In vivo decrease in tumor growth and second engraftment.		[129]
	Breast cancer	In vitro reduction in cell proliferation, tumorsphere formation, and cell viability. In vivo decrease in xenograft tumor growth.		[130]
Fisetin (vegetables and fruits)	Lung cancer	In vitro reduction in cell growth and colony formation.		[132]
	Lung cancer	In vitro reduction in proliferative and invasive capacity via the downregulation of CD44 and CD133 stem-like markers.		[133]

Abbreviations: CSCs, cancer stem cells, HDAC, histone deacetylase, SP, side population, SHH, sonic hedgehog, LGR5, Leucine-rich repeat-containing G-protein coupled receptor 5, EpCAM, epithelial cell adhesion molecule, ALDH, aldehyde dehydrogenases, CXCR4, C-X-C chemokine receptor type 4.

**Table 2 jcm-11-06996-t002:** NPs derived from botanical sources and their effects on CSCs.

Natural Products	Tumor Type	Effects on CSCs	Alone or in Combination with Other Compounds	References
Luteolin (plants)	Breast cancer	In vitro reduction in tumorsphere formation, decrease in stemness marker (ABCG2 and CD44) expression and ALDH1 activity. In vitro reduction in cell viability.	Taxol	[136]
	Prostate cancer	In vitro reduction in tumorsphere formation and of the expression levels of stem-like markers (NANOG, SOX2, and CD44).	Quercetin	[137]
	Oral cancer	In vitro arrest of cell proliferation and migration and decrease in ALDH activity, CD44 expression levels, and IL6/STAT3 axis. In vivo reduction in tumor growth through IL-6/STAT3 signaling inactivation.		[138]
Berberine (plants)	Colorectal cancer	In vitro reduction in tumorsphere formation, cell proliferation, and CD44 and CD133 expression levels. In vivo reduction in tumor growth.	5-FU and irinotecan	[143]
	Pancreatic cancer	In vitro decrease in SP percentage and the expression of the stemness genes (*POU5F1*, *SOX2*, and *NANOG*).	Gemcitabine	[144]
	Ovarian cancer	In vitro reduction in tumorsphere formation, expression of EMT stemness marker expression, and GLI1-BMI1 axis.	Carboplatin and VP-16	[145]
	Neuroblastoma	In vitro downregulation of PI3K/Akt, TGF-β, and MAPK pathways.		[146]
Vincristine (Madagascar periwinkle)	Colorectal cancer	In vitro demethylation of RUNX3.		[153]
Cloroquine (bark of the *Cinchona officinalis* tree)	Breast cancer	In vitro reduction in CD44^+^/CD24^−^ stem-like population.		[155]
	Breast cancer	In vitro reduction in CD44^+^/CD24^−^ and ALDH^+^ stem-like population, impairment of autophagy (increased levels of LCB3-II), and increase in apoptosis and cleaved caspase-3 levels. In vivo reduction in tumor growth and metastatic foci.	Paclitaxel Paclitaxel	[156]
	Breast cancer	In vitro induction of mitochondrial depolarization. In vitro reduction in autophagy and the expression levels of DNA repair machinery proteins.	Carboplatin	[157]
	Pancreatic cancer	In vitro inhibition of CXCL12/CXCR4 axis and SHH pathway. In vivo reduction in the tumorigenic potential of CD133+ subpopulation.		[158]
	Esophageal carcinoma	In vitro reduction in the CXCR4/STAT3 axis.		[159]
Capsaicin (plants of the genus *Capsicum*)	Prostate	In vitro reduction in tumorsphere formation, cell viability, and the expression levels of stem-like markers (CD133, CD44, OCT-4, NANOG, and SOX2) and downregulation of GSK3β/β-catenin pathway.		[161]
Dihydrocapsaicin (derivate of capsaicin)	Brain	In vitro decrease in CD133^+^ subpopulation and induction of cell death.		[164]

Abbreviations: NPs, natural products, CSCs, cancer stem cells, ABCG2, ATP binding cassette subfamily G member 2, ALDH, aldehyde dehydrogenases IL-6, interleukine- 6, SP, side population, POU5F1, POU Class 5 Homeobox 1, TGF-β, tumor growth factor- β, CXCR4, C-X-C chemokine receptor type 4, LCB3-II, microtubule associated protein 1 light chain 3 beta, CXCL12, C-X-C Motif Chemokine Ligand 12.

**Table 3 jcm-11-06996-t003:** NPs derived from marine sources and their effects on CSCs.

Natural Products (Source)	Tumor Type	Effects on CSCs	Alone or in Combination with Other Compounds	References
Alkaloids				
Nortopsentin (deep-sea sponges, Spongsoritesruetzleri)	Colorectal	In vitro arrest of cell proliferation (inhibition of CDK1 activity), induction of apoptosis (Caspase 3), and decrease in stem cell markers (CD44v6) and pathways (Wnt/β-catenin).	Rabusertib	[168]
Renieramycin M (blue sponge *Xestospongia* species)	Non-small-cell lung cancer	In vitro reduction in tumorsphere formation andstem-like markers (CD133, CD44, ALDH1A1).		[169]
Carotenoids				
Fucoxanthinol (brown algae)	Colorectal cancer	In vitro reduction in tumorsphere formation by the inactivation of AKT signaling and the downregulation of PPARβ/δ and PPARγ protein expression, in vitro induction of apoptosis via the reduction in cellular adhesion molecule expression.		[171]
	Colorectal cancer	In vitro decrease in proliferation pathways (JAK/STAT, PI3K/Akt, MAPK, NF-κB).		[172]
Macrolides				
Bryostatin-1 (*Bugula neritina*)	Melanoma cells	In vitro reduction in proliferation and ABCB5^+^ subpopulation.		[176]
	Leukemia	In vitroinduction of apoptosis and reduction in CD34^+^ cell fraction	Gleevec	[177]

Abbreviations: NPs, natural products, CSCs, cancer stem cells, CDK1, Cyclin Dependent Kinase 1, ALDH, aldehyde dehydrogenases, PPARβ/δ, peroxisome-proliferator-activated receptor β/δ, PPARγ peroxisome-proliferator-activated receptor γ.

**Table 4 jcm-11-06996-t004:** Natural products in clinical trials for cancer treatment.

NPs (Source)	Tumor Type	Phase	Patient Number	Parameters	Results	References and CT Number
Curcumin (*Curcuma longa*)	Myeloma	I	15	Patients displaying intolerance to dexamethasone treated with curcumin (3.0–4.0 g/day oral administration) plus immunomodulatory drugs (IMD, lenalidomide) or proteasome inhibitors (PI, bortezomib) for about 6 years.	Curcumin in combination with IMD or PI lessened paraprotein (38%) and plasmacytosis (59%) levels; 12 out of 15 patients were stable.	[189]
	Myeloma and prostate	II	40	Patients were treated with curcumin (4 g) plus piperidine (5 mg) by oral administration for 12 months.	No posted results. First results will be posted after May 2023.	NCT04731844
	Myeloma	IIa	33	The treated patient group (17) was treated with curcumin (8 g/day) for 28 days plus melphalan (4 mg/m^2^) and prednisone (40 mg/m^2^) for 7 days. Control patient group (16) was treated with melphalan, prednisone, and a placebo. The two groups received 4 cycles of treatment.	The treated group displayed an increased overall remission with a reduction in IL-6, VEGF, and TNF-α levels compared with the control group.	[190] ISRCTN14131419
	Metastatic breast cancer	II	150	Treated patient group was treated with intravenous curcumin (300 mg) and paclitaxel (80 mg/m^2^) weekly for 12 weeks. Control patient groups were treated with paclitaxel and a placebo.	Tumor reduction by 50.7% in curcumin treatment compared with 33.3% placebo.	[191] NCT03072992
	Obese women characterized by high risk of developing breast cancer	I	29	The participants received curcumin (50 or 100 mg) by oral administration twice daily for 3 months.	No posted results.	NCT01975363
	Breast cancer patients before surgery	Not applicable	30	Patients were treated with 8 g per day by oral administration for two to four weeks before surgery.	No posted results.	NCT03847623
	Breast cancer patients treated with radiotherapy	II/III	686	Patients were treated with curcumin (6.0 g) by oral administration daily for the entire period of the radiation treatments plus another week.	Breast cancer patients treated with curcumin displayed a reduced dermatitis severity.	NCT01246973
	Metastatic colorectal cancer	IIa	41	Treated patient group was treated with curcumin (2 g/day) orally administered plus standard chemotherapy (FOLFOX) every 2 weeks for 12 cycles.	The clinical trial assessed the safety and effect of curcumin in combination with FOLFOX-based chemotherapy in metastatic colorectal cancer patients.	[192] NCT01490996
EGCG	Ovarian cancer	II	300	Five treatment arms: (i) Combined treatment (TP: paclitaxel 175 mg/m^2^ plus cisplatin 75–100 mg/m^2^ by intravenous administration, or TC: paclitaxel 175 mg/m^2^ plus carboplatin AUC 5 by intravenous administration, plus surgery plus postoperative TP or TC regimen) in combination with I3C (200 mg/day) continuously; (ii) combined treatment plus I3C (200 mg/day) and EGCG (200 mg/day) continuously; (iii) combined treatment plus I3C and EGCG continuously and TP therapy; (iv) combined treatment without TP or TC postoperative regimen; (v) combined treatment.	Patients treated with I3C and EGCG plus chemotherapy showed an increased median overall survival (60 months) and median progression-free survival (48.5 months) compared with single treatment and chemotherapy alone. Moreover, I3C and EGCG treatment reduced cancer recurrence.	[193] ACTRN12616000394448
	Lung cancer	II	83	Prophylactic EGCG group: EGCG (440 umol/L) 0.9% saline solution 3 times/day at the beginning of radiotherapy treatment; therapeutic EGCG group: EGCG 0.9% saline solution 3 times/day in presence of grade 1 esophagitis radiotherapy side effects; conventional therapy group: mLDG (lidocaine 0.16 mg/mL, dexamethasone 0.02 mg/mL, and gentamycin 0.16 mg/mL) 3 times/day in presence of grade 1 esophagitis radiotherapy side effects.	Compared to standard therapies, EGCG preventive treatment and EGCG administration in radiation-treated patients reduced the severity of esophagitis and the levels of pro-inflammatory factors in the serum.	[194] NCT02577393
	Esophageal Cancer	I	15	Six escalating doses (880 umol/L–4400 umol/L) of EGCG weredissolved in 0.9% saline solution and administered three times a day. EGCG solution was given continuously for 8 days before anti-tumor treatment.	No posted results (ongoing trial).	[195] NCT05039983
EGCG (Polyphenon E)	Bladder cancer	II	31	An amount of 800–1200 mg/day of orally administered EGCG for 14–28 days prior to surgery.	The dose-dependent downregulation of two tumor biomarkers, clusterin (an apoptosis marker) and PCNA (a proliferation marker), was seen in EGCG-treated patients, supporting the compound’s chemoprotective activity and use as a neoadjuvant therapy before transurethral resection of bladder tumor or cystectomy, despite the fact that there are no differences in EGCG tumor levels between EGCG-treated and placebo patient groups.	[196]
	Breast cancer	II	32	Subjects are asked to take 4 polyphenol E (200 mg) capsules daily with a meal for the duration of the study. Biomarkers are measured at baseline and then again at presurgery, the end-point for the study within a time frame between 4 and 6 weeks.	Polyphenon E neoadjuvant daily administration induced a decreasein serum HGF levels, without altering those of VEGF.	NCT00676793
	Breast cancer	Ib	40	After completing adjuvant therapy, women with stage I–III breast cancer were randomized to receive Poly E at dosages of 400, 600, or 800 mg twice daily for six months, or a placebo. Samples of blood and urine were collected at the beginning, 2, 4, and 6 months.	After completing adjuvant therapy, women with stage I–III breast cancer were randomized to receive Poly E at dosages of 400, 600, or 800 mg twice daily for six months, or a placebo. Samples of blood and urine were collected at the beginning, 2, 4, and 6 months.	[197]
Resveratrol	Breast cancer	I	39	Resveratrol (5–50 mg) was orally administered for 3 months, twice a day.	After resveratrol treatment, no significant changes in p16, CCND2, APC, and RASSF-1α DNA methylation wereobserved, but only a decreased methylation profile of RASSF-1α and an increase inthe APC profile.	[199]
	Prostate cancer	I	14	Patients were treated with 500 to 4000 mg of muscadine grape extract (MPX, containing 1.2 mg of ellagic acid, 9.2 g of quercetin, and 4.4 g of trans-resveratrol) per day, taken orally for 28 days, with a follow-up of >2 years.	No tolerability problems were observed in patients, and the treatment was deemed to be safe. Additionally, it demonstrated a delay in the recurrence process by extending the PSA doubling time (PSADT) by 5.3 months.	[200]
	Prostate cancer	I	112	Following stratification based on their initial PSADT and Gleason scores, the participants were randomly allocated 1:2:2 to receive a placebo, 500 mg of MPX (low), or 4000 mg of MPX (high), daily.	There was no discernible difference between the control and MPX-treated cohorts in the time it took for PSA to double.	[201] NCT00256334
	Colorectal cancer	I	8	Resveratrol was administered at 20 and 80 mg/day. Resveratrol-containing freeze-dried grape powder (GP) was administered at 0.073 and 0.114 mg/day. Both treatments were taken orally.	Resveratrol/GP treatment significantly inhibited Wnt target gene expression (myc, jun, TCF7, cyclin D1, axin II) in healthy colonic mucosa without effects on tumor mucosa.	[202]
	Colorectal cancer	I	9	A 5.0 g daily dose for 14 days of SRT501 (micronized form of resveratrol), was administered to patients with colorectal cancer and hepatic metastases who were scheduled to undergo hepatectomy.	This treatment method increased drug availability and absorption. After 1–2 weeks of therapy with resveratrol or SRT501, the observed amounts of parent resveratrol and its primary metabolites in the colon tissue of patients were comparable to the efficacious doses of resveratrol utilized in pre-clinical investigations. In addition, cleaved caspase-3, an indication of death, dramatically increased in malignant hepatic tissue after SRT501 therapy by 39% in comparison to tissue from patients who received a placebo.	[203]
	Colorectal cancer	I	20	Colorectal cancer patients were treated daily with resveratrol (0.5 or 1.0 g) for 8 days before surgery.	Treatment reduced ki-67 levels.	[204] NCT00433576
Vincristine	Low-grade glioma	I	497	A total of 497 patients were randomized to receive vincristine carboplatin (VC, vincristine 1.5 mg/m^2^ × 10 weekly and carboplatin 550 mg/m^2^ q 3 weekly) (n = 249) or VC plus etoposide (VCE, etoposide 100 mg/m^2^, days 1, 2,and 3).	The high rates of non-progression after 24 weeks support the use of VC as a first-line treatment.	[210] European Union Clinical Trials Register No. 2005-005377-29
	Diffuse large B-cell lymphoma	II	120	Patients underwent a pre-treatment phase consisting of oral prednisone (60 mg total dosage commencing 1 week before cycle 1, for 4 days (day7 to day4)) and oral vincristine before the first cycle of the ofatumumab (1000 mg every 3 weeks) + miniCHOP regimen (400 mg/m^2^ of intravenous cyclophosphamide, 25 mg/m^2^ of intravenous doxorubicin, 1 mg/m^2^ of intravenous vincristine on day 1 of each cycle, and 40 mg/m^2^ of oral prednisone every day from days 1 to 5).	The pretreatment with ofatumumab and miniCHOP in 80-year-old patients improved overall survival in comparison with standard therapy.	[211] NCT001195714
Bryostatin-1	Renal cell carcinoma	II	30	Patients were treated for 30 min with an intravenous infusion of bryostatin-1 (25 microg/m^2^) with formulation PET (polyethyleneglycol, ethanol, and Tween 80) on days 1, 8, and 15 of each 28-day cycle.	Patients responded to the treatment without severe adverse effects.	[213]
	Low-grade non-Hodgkin lymphoma and chronic lymphocytic leukemia	II	25	Patients were treated for 72 h with a continuous infusion of bryostatin-1 (120 microg/m^2^) per course every 2 weeks immediately followed by vincristine (from 0.5 mg/m^2^ to 2 mg/m^2^) administration by bolus i.v. injection.	Treatment with bryostatin-1 resulted in one patient in complete remission and two in partial remission. Moreover, this treatment promoted a differentiative state of CLL cells, demonstrated bythe presence of CD11c/CD22/CD20 B-cell subpopulations.	[214]
Sulforaphane	Prostate cancer	nd	98	Patients were treated with BSE (200 µmol daily) or a placebo for 4–8 weeks.	Forty differently expressed genes linked to BSE treatment, including the downregulation of two prostate cancer-related genes. Supplementing with BSE is associated with alterations in gene expression but not with changes in prostate tissue biomarkers.	[215] (NCT01265953)
	Prostate cancer	II	61	Patients were given a weekly 300 mL serving of soup produced from either regular broccoli (the control) or one of two experimental broccoli genotypes with increased glucoraphanin concentrations that delivered 3 or 7 times the amount of the control, respectively.	Downregulation of genes linked to inflammation processes and epithelial–mesenchymal transition in a dose-dependent manner in glucoraphanin-rich broccoli-soup-consuming men. An inverse relationship was found between the consumption of cruciferous vegetables and a decreased risk of prostate cancer advancement in males under active monitoring.	[216] (NCT01950143)
Fisetin	Colorectal cancer	I	38	CRC patients treated with chemotherapy were randomized to receive either 100 mg fisetin (n = 18) or placebo (n = 19) for 7 weeks.	After fisetin administration to CRC patients, plasma levels of IL-8 and hs-CRP dropped dramatically as a lower expression of MMP-7.	[217] (code: IRCT2015110511288N9)

Abbreviations: IMD, immunomodulatory drugs, PI, proteasome inhibitor, VEGF, vascular endothelial growth factor, IL-6, interleukine-6, TNF-α, tumor necrosis factor-α, FOLFOX, leucovorin calcium (folinic acid), fluorouracil, and oxaliplatin., EGCG, epigallocatechin-3-gallate, TC, taxotere and cyclophosphamide, TP, docetaxel plus cisplatin, PCNA, proliferating cell nuclear antigen, HGF, hepatocyte growth factor, CCND2, Cyclin D2, APC, WNT signaling pathway regulator, RASSF-1α, Ras association domain-containing protein 1 I, TCF7, Transcription Factor 7, SRT501, micronized form of resveratrol, miniCHOP, mini-cyclophosphamide, doxorubicin, vincristine, and prednisone, CLL, chronic lymphocytic leukemia BSE, broccoli sprout extract, CRC, colorectal cancer.

## Data Availability

Not applicable.

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
