# Peer review of "Destroying the Shield of Cancer Stem Cells: Natural Compounds as Promising Players in Cancer Therapy"

_jcm, 2022, doi:10.3390/jcm11236996_

Round 1

Reviewer 1 Report

Lo Iacono et al. summarize a review in their manuscript of various groups of natural compounds targeting cancer stem cells for adjuvant therapy in cancer treatments.

The manuscript is well-written and details a novel and timely research field in oncology.

`due to rising drug costs and the need to protect the environment and give ecological credentials to chemotherapy compounds` I understand your point. However, the main reason in my opinion is that FDA-approved or natural drugs get through drug development phases much quicker and cheaper (with less side-effects), therefore patient treatments can start much sooner compared to other chemo drug developments. Could you please include these thoughts as well (including the abstract as well)?

Figure 1. is more for an academic presentation rather than for publication purposes. Please, modify the figure to a more suitable style. Resolution of Figure 1 is low, maybe because it has been inserted into the pdf file for reviewers. I suggest using the word `increased` instead of `high` (expression) in the figure. Please consider adding `Changed metabolism` (`CSC metabolic reprogramming`).

I suggest adding a table, if space allows, to highlight the NPs currently used in clinical trials (that would improve the value of the manuscript as well).

`reducing dangerous food uptake` (line 203), Do you mean unhealthy processed meats? I suggest using a different phrase.  

Taxus Brevifolia (line 335) please correct to Taxus brevifolia.

Other compounds that can be added to the list: Sulphoraphane, Diosmin, fisetin.

English language mistakes need to be addressed.

Author Response

Reviewer 1

Lo Iacono et al. summarize a review in their manuscript of various groups of natural compounds targeting cancer stem cells for adjuvant therapy in cancer treatments.

The manuscript is well-written and details a novel and timely research field in oncology.

We thank Reviewer #1 for his/her valuable comments.

`due to rising drug costs and the need to protect the environment and give ecological credentials to chemotherapy compounds` I understand your point. However, the main reason in my opinion is that FDA-approved or natural drugs get through drug development phases much quicker and cheaper (with less side-effects), therefore patient treatments can start much sooner compared to other chemo drug developments. Could you please include these thoughts as well (including the abstract as well)?

As suggested by the Reviewer, we add a statement in the abstract, in the introduction, and in the conclusion and perspective sections.

Figure 1. is more for an academic presentation rather than for publication purposes. Please, modify the figure to a more suitable style. Resolution of Figure 1 is low, maybe because it has been inserted into the pdf file for reviewers. I suggest using the word `increased` instead of `high` (expression) in the figure. Please consider adding `Changed metabolism` (`CSC metabolic reprogramming`).

On the basis of all the reviewers’ comments, we modify the figures. Specifically, i) we combined Figures 1 and 2 into a single Figure; ii) we better depict that cancer stem cells are a small subpopulation within the tumor mass, mainly composed of differentiated cells, and are refractory to radio- and chemo-therapy; iii) we better indicated that NPs from a different source is able to reduce and impair the principal features of CSCs directly; iv) we included the metabolic reprogramming in the stem-like traits of CSCs. Moreover, in the new version of the manuscript, we increased the image resolution and included a more detailed description of the Figure.

I suggest adding a table, if space allows, to highlight the NPs currently used in clinical trials (that would improve the value of the manuscript as well).

In the new version of the manuscript, the information regarding the use of NPs in clinical trials, including parameters, therapeutic effects and number of clinical trials, are reported in Table 4.

`reducing dangerous food uptake` (line 203), Do you mean unhealthy processed meats? I suggest using a different phrase. 

We apologize for the confusion. In the new version of the manuscript, we removed this statement.

Taxus Brevifolia (line 335) please correct to Taxus brevifolia.

We unified the nomenclature throughout the entire manuscript.

Other compounds that can be added to the list: Sulphoraphane, Diosmin, fisetin.

In the new version of the manuscript, we reported studies regarding Sulphoraphane and fisetin activity in reducing CSC features in the “NP s derived from dietary sources” paragraph. However, the literature regarding the anticancer effect of diosmin did not include studies on CSCs.

English language mistakes need to be addressed.

We corrected typos and grammar errors throughout the manuscript.

Reviewer 2 Report

In view of protection of environment and increase of the ecosystem benefit, Giorgio Stassi et al. focused on the application of natural active ingredients in adjuvant therapy targeting cancer stem cells. This manuscript is well written and comprehensively summarizes the research progress. However, some details should be concerned before publishing.

1. Retinoids should be classified as terpenes according to the type of skeleton.

2. Figure 1 and Figure 2 are very similar. It is suggested to keep one figure or make appropriate modifications.

3. Cancer stem cells is a small part of in cancer cell population. It should be reflected in Figures 1 and 2.

4. Polyphenols and flavonoids are classified in Table 1, but EGCG is also belong to flavonoid. In fact, polyphenols includes flavonoids.

5. The classification of flavonoids in Line288 is worth discussing. Four types of flavonoids have been reported: flavanones, flavones, flavanols, and anthocyanins.

6. The clinical trials of natural product for cancer treatment should be summarized in a table.

7. The Latin names involved in the text should be expressed more standardized.

8. There are too many references. It is recommended to simplify.

9. Some writing norms and multiple errors should be concerned carefully, such as:

1) Line 214, “NF-kb” should be changed into “NF-kB”.

2) Line 256, “(“ should be deleted.

3) Line 380, chloroquine is a monomeric compound. “Among these chloroquine” should be revised correctly.

4) Line54, (PMID: [4-6]) ,”PMID:” Should be deleted.

5) Line66, (PMID: [17], iv) ,”PMID:” Should be deleted.

6) Line88, (10.1016 / j. jca. 2012.04.013), the DOI number should be deleted.

7) Line568 "In another phase I pilot trial (NCT00256334" , the brackets are incomplete.

8) The location of clinical research number and citation number should be unified in the whole text, such as Line532 ([212], NCT02577393), Line533 ([213], NCT05039983) Line527 [211] (ACTRN12616000394448).

9) Line567, "are need"should be changed into "are needed". 

10) Line590, "by inhibition of tubulin polymerization" should be changed into “by inhibiting tubulin polymerization”.

Author Response

Reviewer 2

In view of protection of environment and increase of the ecosystem benefit, Giorgio Stassi et al. focused on the application of natural active ingredients in adjuvant therapy targeting cancer stem cells. This manuscript is well written and comprehensively summarizes the research progress. However, some details should be concerned before publishing.

We thank Reviewer #2 for the valuable comments.

  1. Retinoids should be classified as terpenes according to the type of skeleton.

As suggested by the Reviewer, we classified retinoids as terpenes.

  1. Figure 1 and Figure 2 are very similar. It is suggested to keep one figure or make appropriate modifications.

On the basis of all the reviewers’ comments, we modify the figures. Specifically, i) we combined Figures 1 and 2 into a single Figure; ii) we better depict that cancer stem cells are a small subpopulation within the tumor mass, mainly composed of differentiated cells, and are refractory to radio- and chemo-therapy; iii) we better indicated that NPs from the different source are able to directly reduce and impair the principal features of CSCs; iv) we included the metabolic reprogramming in the stem-like traits of CSCs. Moreover, in the new version of the manuscript, we increased the image resolution and included a more detailed description of the Figure.

  1. Cancer stem cells is a small part of in cancer cell population. It should be reflected in Figures 1 and 2.

Please see answer to the above point 2.

  1. Polyphenols and flavonoids are classified in Table 1, but EGCG is also belong to flavonoid. In fact, polyphenols includes flavonoids.

We agree with this Reviewer and accordingly, we have now changed the Table 1 classification.

  1. The classification of flavonoids in Line288 is worth discussing. Four types of flavonoids have been reported: flavanones, flavones, flavanols, and anthocyanins.

As requested by this Reviewer, we corrected the statement.

  1. The clinical trials of natural product for cancer treatment should be summarized in a table.

In the new version of the manuscript, the information regarding the use of NPs in clinical trials, including parameters, therapeutic effects, and number of clinical trials, are reported in Table 4.

  1. The Latin names involved in the text should be expressed more standardized.

We unified the nomenclature of natural compounds throughout the entire manuscript.

  1. There are too many references. It is recommended to simplify.

As this Reviewer requested, we reduced the number of references.

  1. Some writing norms and multiple errors should be concerned carefully, such as:

1) Line 214, “NF-kb” should be changed into “NF-kB”.

2) Line 256, “(“ should be deleted.

3) Line 380, chloroquine is a monomeric compound. “Among these chloroquine” should be revised correctly.

4) Line54, (PMID: [4-6]) ,”PMID:” Should be deleted.

5) Line66, (PMID: [17], iv) ,”PMID:” Should be deleted.

6) Line88, (10.1016 / j. jca. 2012.04.013), the DOI number should be deleted.

7) Line568 "In another phase I pilot trial (NCT00256334" , the brackets are incomplete.

8) The location of clinical research number and citation number should be unified in the whole text, such as Line532 ([212], NCT02577393), Line533 ([213], NCT05039983) Line527 [211] (ACTRN12616000394448).

9) Line567, "are need"should be changed into "are needed".

10) Line590, "by inhibition of tubulin polymerization" should be changed into “by inhibiting tubulin polymerization”.

We corrected typos and grammar errors throughout the manuscript.

Reviewer 3 Report

Lo Iacono et. al. reviewed natural compounds to target cancer stem cells. They reviewed natural compounds that can inhibit pathways involved in the maintenance of cancer stem cells and natural compounds that can sensitize cancer stem cells to standard chemotherapeutic treatments. The review is very informative and well-described. I have one comment to improve the review:

1-    It will be helpful for the readers if there is a table with the significant natural compounds in clinical studies and a description of the trials.

Author Response

Reviewer 3

Lo Iacono et. al. reviewed natural compounds to target cancer stem cells. They reviewed natural compounds that can inhibit pathways involved in the maintenance of cancer stem cells and natural compounds that can sensitize cancer stem cells to standard chemotherapeutic treatments. The review is very informative and well-described. I have one comment to improve the review:

We thank Reviewer #1 for his/her valuable comments.

1- It will be helpful for the readers if there is a table with the significant natural compounds in clinical studies and a description of the trials.

In the new version of the manuscript, the information regarding the use of NPs in clinical trials, including parameters, therapeutic effects, and the number of clinical trials, are reported in Table 4.

Reviewer 4 Report

Lo Iacono and colleagues present a thorough review on the anti-oncogenic effects of natural compounds (NPs) from different sources on cancer stem cells (CSCs) in the malignant settings. This topic has a paramount impact considering the unresolved resistance of cancer to chemo-radiotherapeutics. The manuscript is clearly written with significant technical insufficiencies.

Specific comments

Title. “Destroying the shield of cancer stem cells: natural compounds as adjuvant treatment. The three major elements of this review are CSCs, NPs and cancer therapy, which should be included in the title. Kindly revise as appropriate.

Abstract and Introduction.  A casual search of PubMed returned a handful of reviews on this topic. Therefore, the authors are encouraged to impart stronger rationale in the Abstract and Introduction why and how this manuscript will advance this field. 

Introduction. Line 35-44. The authors strongly support the concept of Green Oncology model, which is a part of a global challenge with impact on social, environmental, economic including medical landscapes. While it is true that this topic is associated with Green Oncology, this is not a major component of this review, hence, is not essential on this occasion. In general, line 35-44 provide a positive outlook considering the impact of this review on Green Oncology, hence, a good contribution under Conclusions and Perspectives.

Under Introduction, the authors should briefly introduce CSCs, their presence in the different types of cancer and the current notion that this cell population could be the root of drug tolerance in the malignant settings. Brief statement indicating that NPs have proven efficacies as effective adjuvants in cancer therapy is also essential.

Figure1. What relevance do DCs have in this scenario? It is suggested that all the CSC-based drug resistance mechanisms be aligned on the shield because these are the factors that protect them from chemo- and radiotherapy.

3.1. Line 202-204. This statement relating to reduction of tumor development is not vital on this occasion considering the topic of the anti-tumorigenic effects of NPs.

Line 331. The authors should indicate that Table 1 is a summary of the studies reviewed under section 3.1.

 Table1. Scientific names are always italicized. Genus name is capitalized; species is not capitalized. Kindly consider this in the entire manuscript. Table 1, 2, and 3 should include two parameters: tumor type and experimental conditions (such as in vitro or in vivo) to provide a better overview of the data.

Table 1, 2, and 3. Apparently, the authors, in most cases, summarized the reviewed data under “reduction of tumor spheres, arrest of cell proliferation, migration and tumorigeneis, induction of apoptois and authophagy and decrease of stem-like markers and pathways”. To summarize the reviewed data under these headings is inappropriate and not acceptable. The authors are requested to describe the findings as reported by the respective authors. For instance, “Decrease in stem cell markers and pathways”, here, the authors should explicitly indicate which stem cell markers and pathways were implicated.

Several studies were not included in Table 1 such as ref 93-95, line 213-215; ref 106, 138, 133,135 and 150.

It should also be indicated whether the effect of a particular NP was in combination with other compound/chemotherapeutic drugs such as in ref 93-95, ref 141, ref 157 (Table2)

Table 2, results of ref 171, were incomplete; data on line 374-377 were not taken into consideration.

Table 2, ref 180, line 399, this study found that capsaicin hampers CSC survival – kindly justify why these important findings were not included as well as ref 183.

Table 3, results of ref 186 were not included. Reviewed data of ref 187 did not report any effect of NORA 234, an analog of nortopsentin, on autophagy and an interruption of any signaling pathway.  Kindly check.

Figure 2. Line 465 does not describe this figure at all. Kindly include a brief description of this figure. This illustration shows that marine, food and botanical compounds support chemo-radiotherapy to interrupt the properties of CSCs and implicated pathways, thus, breaking the shield and sensitize CSCs to cancer treatment. However, in the reviewed literatures, unless otherwise stated, the natural compounds directly attenuate or even totally interrupt the properties of CSCs and signaling pathways resulting to anti-oncogenic effects. Kindly revise this illustration as appropriate.

4. Natural products in clinical trial for cancer treatment. To provide an overview of the different clinical trials for each compound, a summary in tabularized form is essential. Kindly include the parameters, therapeutic effect and clinical trial for each NP.

Conclusions. The current form of conclusions is more of expanding the future perspectives considering the role of natural products in cancer therapy. In this section, the authors should draw a brief summary (based on reviewed literatures) backing up the claim that NPs are able to destroy the shield of CSCs.

Author Response

Reviewer 4

Lo Iacono and colleagues present a thorough review on the anti-oncogenic effects of natural compounds (NPs) from different sources on cancer stem cells (CSCs) in the malignant settings. This topic has a paramount impact considering the unresolved resistance of cancer to chemo-radiotherapeutics. The manuscript is clearly written with significant technical insufficiencies.

We thank Reviewer #4 for the constructive comments. In the revised version of the manuscript, we better organized Tables adding more information on the effect of NPs on CSC features and we improve the quality of the figure to address your concerns.

Specific comments

  1. Title. “Destroying the shield of cancer stem cells: natural compounds as adjuvant treatment. The three major elements of this review are CSCs, NPs and cancer therapy, which should be included in the title. Kindly revise as appropriate.

On the basis of reviewer’s comment, we modified the title accordingly: Destroying the shield of cancer stem cells: natural compounds as promising players in cancer therapy

  1. Abstract and Introduction.  A casual search of PubMed returned a handful of reviews on this topic. Therefore, the authors are encouraged to impart stronger rationale in the Abstract and Introduction why and how this manuscript will advance this field. 

In our manuscript, we deepened our knowledge about the use of major natural compounds derived from food, botanical and marine sources in reducing the stem-like features of cancer stem cells and their use in clinical trials. In the new version of manuscript, we pointed out these relevant findings in the abstract and the introduction section.

  1. Introduction. Line 35-44. The authors strongly support the concept of Green Oncology model, which is a part of a global challenge with impact on social, environmental, economic including medical landscapes. While it is true that this topic is associated with Green Oncology, this is not a major component of this review, hence, is not essential on this occasion. In general, line 35-44 provide a positive outlook considering the impact of this review on Green Oncology, hence, a good contribution under Conclusions and Perspectives.

We agree with this Reviewer and accordingly, we moved the Green Oncology concept in the conclusion and perspective section.

  1. Under Introduction, the authors should briefly introduce CSCs, their presence in the different types of cancer and the current notion that this cell population could be the root of drug tolerance in the malignant settings. Brief statement indicating that NPs have proven efficacies as effective adjuvants in cancer therapy is also essential.

As requested by this Reviewer, we better described the role of cancer stem cells in radio-/chemo-therapy resistance and the use of natural products in adjuvant settings in the introduction section.

  1. Figure1. What relevance do DCs have in this scenario? It is suggested that all the CSC-based drug resistance mechanisms be aligned on the shield because these are the factors that protect them from chemo- and radiotherapy.

On the basis of all the reviewers’ comments, we modify the figures. Specifically, i) we combined Figures 1 and 2 into a single Figure; ii) we better depict that cancer stem cells are a small subpopulation within the tumor mass, mainly composed of differentiated cells, and are refractory to radio- and chemo-therapy; iii) we better indicated that NPs from the different source are able to reduce and impair the principal features of CSCs directly; iv) we included the metabolic reprogramming in the stem-like traits of CSCs. Moreover, in the new version of the manuscript, we increased the image resolution and included a more detailed description of the Figure.

  1. Line 202-204. This statement relating to reduction of tumor development is not vital on this occasion considering the topic of the anti-tumorigenic effects of NPs.

As suggested by the Reviewer, in the new version of the manuscript we removed this statement.

  1. Line 331. The authors should indicate that Table 1 is a summary of the studies reviewed under section 3.1.

We have now reported this information in the text.

  1. Table1. Scientific names are always italicized. Genus name is capitalized; species is not capitalized. Kindly consider this in the entire manuscript. Table 1, 2, and 3 should include two parameters: tumor type and experimental conditions(such as in vitro or in vivo) to provide a better overview of the data.

Table 1, 2, and 3. Apparently, the authors, in most cases, summarized the reviewed data under “reduction of tumor spheres, arrest of cell proliferation, migration and tumorigeneis, induction of apoptois and authophagy and decrease of stem-like markers and pathways”. To summarize the reviewed data under these headings is inappropriate and not acceptable. The authors are requested to describe the findings as reported by the respective authors. For instance, “Decrease in stem cell markers and pathways”, here, the authors should explicitly indicate which stem cell markers and pathways were implicated.

As this Reviewer requested, we modified Tables 1, 2, and 3 taking into account all the concerns. Specifically, we included information regarding tumor types and the use of natural compounds alone or in combination with other products or drugs, and we better specified the effects of natural compounds on CSCs, highlighting the experimental conditions.

  1. Several studies were not included in Table 1 such as ref 93-95, line 213-215; ref 106, 138, 133,135 and 150.

In Tables 1-3 we summarized the effects of NP treatment regarding the reduction of CSC features, such as a decrease of sphere-forming, tumorigenic potential, and expression levels of stem-like markers. Accordingly, we did not report studies that described the main anticancer effects of NPs on differentiated cancer cells.

  1. It should also be indicated whether the effect of a particular NP was in combination with other compound/chemotherapeutic drugs such as in ref 93-95, ref 141, ref 157 (Table2)

Please see answer to the above point 8.

  1. Table 2, results of ref 171, were incomplete; data on line 374-377 were not taken into consideration.

As requested by this Reviewer, we added in Table 2 the missing data regarding the results described in ref 171.

Table 2, ref 180, line 399, this study found that capsaicin hampers CSC survival – kindly justify why these important findings were not included as well as ref 183.

In the new Table 2, we reported the findings about the capacity of capsaicin to reduce CSC viability.

  1. Table 3, results of ref 186 were not included. Reviewed data of ref 187 did not report any effect of NORA 234, an analog of nortopsentin, on autophagy and an interruption of any signaling pathway.  Kindly check.

We did not include reference 186 in Table 3 causes the reported findings were obtained in differentiated cancer cells. In colorectal CSCs, Di Franco et al. demonstrated that NORA234 treatment, despite an initial reduction of CSC proliferative capacity, increases the percentage of CD44v6+/Wnthigh CSC subpopulation. NORA234 in combination with rabusertib, a Check1 inhibitor, induced apopotosis with a concomitant reduction of the CD446+/Wnthigh CSCs. These findings were reported in the new Table 3.

  1. Figure 2. Line 465 does not describe this figure at all. Kindly include a brief description of this figure. This illustration shows that marine, food and botanical compounds support chemo-radiotherapy to interrupt the properties of CSCs and implicated pathways, thus, breaking the shield and sensitize CSCs to cancer treatment. However, in the reviewed literatures, unless otherwise stated, the natural compounds directly attenuate or even totally interrupt the properties of CSCs and signaling pathways resulting to anti-oncogenic effects. Kindly revise this illustration as appropriate.

Please see the answer to the above point 5.

  1. Natural products in clinical trial for cancer treatment. To provide an overview of the different clinical trials for each compound, a summary in tabularized form is essential. Kindly include the parameters, therapeutic effect and clinical trial for each NP.

In the new version of the manuscript, the information regarding the use of NPs in clinical trials, including parameters, therapeutic effects and number of clinical trials, are reported in the Table 4.

  1. Conclusions. The current form of conclusions is more of expanding the future perspectives considering the role of natural products in cancer therapy. In this section, the authors should draw a brief summary (based on reviewed literatures) backing up the claim that NPs are able to destroy the shield of CSCs.

On the basis of all the reviewers’ comments, we highlighted the in vitro anti-tumor activity of NPs in the reduction of CSC features in the conclusion and perspective section.

Round 2

Reviewer 1 Report

Please, differentiate the labels in Figure 1. My concern is that the labels are the same on both sides of the image but you are saying that natural compounds are affecting CSCs. 

If you re showing that natural compounds are weakening CSCs, than you might need to show that High survival will change to Lower survival, stemness induction to reduced stemness, ROS decrease - to changed ROS level. 

In the figure legend I suggest to replace the expression `Lessen` to reduce. 

Author Response

Please, differentiate the labels in Figure 1. My concern is that the labels are the same on both sides of the image, but you are saying that natural compounds are affecting CSCs. 

If you are showing that natural compounds are weakening CSCs, then you might need to show that High survival will change to Lower survival, stemness induction to reduced stemness, ROS decrease - to changed ROS level. 

In the figure legend, I suggest replacing the expression `Lessen` to reduce. 

We thank Reviewer #1 for his/her valuable comments. We differentiated the labels in Figure1, and we modified the expression in all gates, to highlight that natural compounds can weak CSCs.

In the figure legend, we substitute the term “Lessen” with “to reduce”

Reviewer 4 Report

The authors have addressed the Reviewer`s comments appropriately.

Author Response

We thank Reviewer#4